# Point Cloud Densification Algorithm for Multiple Cameras and Lidars Data Fusion

**DOI:** 10.3390/s24175786

**Published:** 2024-09-05

**Authors:** Jakub Winter, Robert Nowak

**Affiliations:** Institute of Computer Science, Warsaw University of Technology, Nowowiejska 15/19, 00-665 Warsaw, Poland

**Keywords:** data fusion, point cloud densification, autonomous vehicle perception systems, stereo vision, lidar, camera, dynamic programming, open source, C++, Python 3

## Abstract

Fusing data from many sources helps to achieve improved analysis and results. In this work, we present a new algorithm to fuse data from multiple cameras with data from multiple lidars. This algorithm was developed to increase the sensitivity and specificity of autonomous vehicle perception systems, where the most accurate sensors measuring the vehicle’s surroundings are cameras and lidar devices. Perception systems based on data from one type of sensor do not use complete information and have lower quality. The camera provides two-dimensional images; lidar produces three-dimensional point clouds. We developed a method for matching pixels on a pair of stereoscopic images using dynamic programming inspired by an algorithm to match sequences of amino acids used in bioinformatics. We improve the quality of the basic algorithm using additional data from edge detectors. Furthermore, we also improve the algorithm performance by reducing the size of matched pixels determined by available car speeds. We perform point cloud densification in the final step of our method, fusing lidar output data with stereo vision output. We implemented our algorithm in C++ with Python API, and we provided the open-source library named Stereo PCD. This library very efficiently fuses data from multiple cameras and multiple lidars. In the article, we present the results of our approach to benchmark databases in terms of quality and performance. We compare our algorithm with other popular methods.

## 1. Introduction

The fused data allow a more accurate analysis due to the usage of a complete, multidimensional description of the world. Combining data from various sources became more and more popular both due to the growing number of sensors and the increasing computing power of computers. Fusing multiple camera images with lidar point clouds has applications in augmented reality, robotics and autonomous driving. This process increases the quality of object detection and object classification in terms of both sensitivity and specificity.

The camera produces two-dimensional (2D) images. It is a dense, regular grid of pixels with a particular color.

Due to progress in digital camera construction (using CCD or CMOS sensors), images contain a very large number of such pixels, for example, 1920×1080≈2 million pixels. The typical color depth is almost 24 bits = ≈16 million.

Three-dimensional (3D) lidar scanner, on the other hand, produces point clouds in three-dimensional space. A point cloud consists of a number of points described by the three coordinates *x*, *y* and *z* representing the location of a point and possible attributes of that point, such as the intensity of light reflected by the object. Lidar point clouds do not contain information about the entire space and may have an irregular structure. Depending on the equipment used and the space being observed, point clouds points may contain a varying number of points, but this number is far smaller than the number of pixels in the images, typically around 100,000 points. In addition, the point cloud becomes sparser as the distance from the sensor increases, so that an object far from the sensor may not be visible in the cloud.

The problem of fusing camera images with 3D lidar points has become important and widely discussed in the literature since the development of such sensors [1,2,3]. The fusion needs a common coordinate system for all fused sensors. In the literature, three approaches are the most popular [4]:Point cloud densification—creation of point clouds based on pairs of stereo vision images and camera calibration data, then combining point clouds;Coloring of lidar point cloud based using colors from camera images;Projection of 3D lidar data on 2D, then fusing 2D images.

The most promising method is point cloud densification, i.e., adding new points obtained by reconstructing three-dimensional space to the existing lidar point cloud [5]. It was applied in moving platforms (aircraft, boat, automobile) [6] and has many applications in geodesy, cartography, environmental monitoring, object detection and others.

Combining several point clouds, e.g., from multiple lidars, is a fairly straightforward operation; it basically involves creating a new point cloud containing points from several clouds, remembering to first transform them to the same coordinate system by multiplying the matrix containing the points by an extrinsic calibration matrix describing rotation and translation.

Indeed, a more difficult problem is to obtain a point cloud from camera data. The problem becomes easier for a pair of stereoscopic images. Obtaining a point cloud from such a pair of images requires finding a pixel match between the images and calculating a disparity map. There are several methods used to solve the pixel-matching problem, for example based on an absolute difference sum or mutual correlation or global optimization of the cost function [7].

Matching pixels in a pair of images can also be found with the Needleman–Wunsch algorithm [8]. This is an algorithm based on dynamic programming, which was originally used for amino-acid or nucleic-acid sequence alignment. In our work, we adapt it to the problem of finding matching pixels and then calculating the coordinates of three-dimensional points. Such a technique was also mentioned in review [9].

Several studies have proposed similar solutions to generate a densification of lidar data [10]. Some methods have proposed cooperative [11,12], centralized [13], decentralized [14], or grid-based algorithms [15] to merge two or more of these sensors: lidar, stereo-vision camera [16], mono camera [17], event camera [18], and also IR camera [19].

There are open libraries and tools allowing the processing of camera and lidar data. The most popular are outlined below:Point Cloud Library (PCL) [20]—a popular library for point cloud processing, PCL is a free and open-source solution. Its functionalities are focused on laser scanner data, although it also contains modules for processing stereo vision data. PCL is a C++ language library, although unofficial Python 3 language bindings are also available on the web, e.g., [21], which allows you to use some of its functionality from within the Python language.OpenCV [22]—one of the most popular open libraries for processing and extracting data from images. It also includes algorithms for estimating the shapes of objects in two and three dimensions from images from one or multiple cameras and algorithms for determining the disparity map from stereo vision images and 3D scene reconstruction. OpenCV is a C++ library with Python bindings.

We propose a new point cloud densification algorithm for multiple cameras and lidars data fusion using point cloud densification. We improved the dynamic programming algorithm performance by reducing the size of matched pixels determined by available car speeds, implementing it in C++ using parallel computing available on modern computer platforms. Furthermore, we developed a new library containing this algorithm, providing a solution that is fully and easily accessible from the Python language. In addition, we also cared about the best possible quality results by using the affine gap penalty function and filling the not-matched pixels with the colors of their neighbours. Our tool is mainly intended to increase the sensitivity and specificity of autonomous vehicle perception systems.

Currently, deep artificial neural networks are replacing the whole processing pipeline [23]; however, the performance in such systems is an issue, as well as the output for non-standard images, e.g., images not presented in the training dataset. This issue is important in critical systems, such as the perception systems of autonomous vehicles.

In the remainder of the paper, we discuss our fusion method in detail, particularly the proposed stereo-matching algorithm inspired by the algorithm used in bioinformatics and how to obtain the final disparity map based on the obtained matching. We describe the new methods used to improve the quality and the performance. In Section 3, we discuss the results of the proposed algorithm, present the benchmark datasets used, and compare the results of our algorithm with other popular methods. Finally, in Section 4 and Section 5, we summarize information about the implemented library, and we also discuss the possible applications and further research directions.

## 2. Materials and Methods

The presented approach is the realization of our point cloud densification idea. A new method for stereo vision is proposed; then, point clouds are combined by transformation to the same coordinate system, as depicted in Figure 1.

Data fusion is the concatenation (merging) of individual points from many point clouds of two kinds: lidar output or our stereovision algorithm output. The point clouds from multiple lidars and multiple camera pairs are then combined into a single point cloud, having first been transformed into a common coordinate system. In this way, the single point cloud is obtained containing points from lidar and color points from camera images. This process has linear time complexity.

### 2.1. New Algorithm for Stereo Vision

Assuming a pinhole camera model and projective geometry (extension to Euclidean geometry where points of an intersection of lines are considered), moreover, considering two cameras with the same focal length and placed in such a way that their planes are parallel, we can show that the pixels corresponding to the same point in 3D are in the same row in both images; they have the same *y* coordinate value [24].

Based on the disparity and camera parameters, the distance of the objects from the camera can be calculated, revealing the coordinates of the 3D points. A projection matrix, which is a composite of the camera’s intrinsic matrix—describing its intrinsic parameters—and an extrinsic calibration matrix, the same as for lidar, for example, describing rotation and translation, is used to obtain points in the appropriate coordinate system. Pixels with calculated depth are multiplied by this matrix to obtain 3D points.

Dynamic programming is often used to solve optimization problems. It involves dividing a complex problem into many smaller subproblems [25]. These subproblems are not independent of each other but are solved sequentially. The key to designing a dynamic programming algorithm is to find a suitable recursive equation.

When matching pixels on a pair of stereo vision images, the idea is to find the best match, such as the best pixel match in each line of images. By using a dynamic programming algorithm, similar pixels can be matched, and non-matching pixels can be skipped. For this purpose, we adapted the Needleman–Wunsch algorithm (dynamic programming) to the pixel sequence-matching problem.

The need for pixel skipping is due to the fact that we want to match one pixel from one line with one pixel from the other line, and the fact that there are situations when a given fragment of space is visible only in one image, because at different camera positions, different fragments of space are invisible: for example, they are obscured by an object closer to the camera. In such a situation, the pixel representing the color of such a fragment of space should not be matched with any pixel from the other image. However, the problem of matching pixels requires the use of a different matching function and a different gap penalty function than in the original version of the algorithm other than a constant.

In each step of the algorithm, the value of the recursive equation is calculated for the corresponding arguments by reading the previously calculated values from an auxiliary matrix, the cells of which are completed in subsequent steps of the algorithm.

A measure of the alignment of pixels is a measure of their similarity. In our solution, we used the sum of the absolute differences of the color channels in the color space as the similarity function. The color similarity function in RGB space is represented by Equation (Equation 1), where e(i,j) is the similarity of the *i*-th pixel from the left image with the *j*-th pixel from the right image, $e_{max}$ is the highest possible similarity (set as a parameter), P(i) is the value of the pixel’s color channel, and the letters in the subscripts denote the channel (*R*, *G* or *B*) and the image (left or right) from which the value is read.
(1)$$e\left(i,j\right)=e_{max}-\left(\right|P_{R,l}\left(i\right)-P_{R,r}\left(j\right)|\;+\;|P_{G,l}\left(i\right)-P_{G,r}\left(j\right)|\;+\;|P_{B,l}\left(i\right)-P_{B,r}\left(j\right)\left|\right)$$

This function can take both positive values, meaning a lot of similarity, and negative values, meaning a lot of difference.

The algorithm does not require that the images be saved using a specific color space; it can be RGB, YUV or grayscale, for example. Because the Needleman–Wunsch algorithm assumes that the matching function returns positive values (indicating a match) and negative values (indicating no match), the parameter of our algorithm is the reward for a perfect match—a match of pixels with identical color. This is a value added to the calculated color difference. The values of penalties and rewards given as parameters of our algorithm should have an absolute value less than 1; in a further step, they are scaled accordingly.

For each pair of pixels, the most beneficial action is calculated. The possible actions for a pair of pixels are to associate them with each other or to insert a gap (hole) on one line, i.e., to skip a pixel. The choice of action is influenced by the values in the neighboring cells of the auxiliary matrix, the alignment of the pixels and the cost of performing the gap. The cost of a gap in our solution is interval-constant (a different penalty for starting a gap and a different penalty for continuing a gap—omitting the next neighboring pixel on one of the lines). The penalty for the *n*-th pixel gap was defined by Equation (Equation 2), where $d_{start}$ and $d_{continue}$ are the parameters of the algorithm.
(2)$$d\left(n\right)=\left\{\begin{array}{cc} d_{start} & n=0\\ d_{continue} & n=1\\ 0 & n>1 \end{array}\right.$$

The choice of such a gap penalty resulted from the observation that the points to which neighboring pixels correspond are close to each other—they belong to the same object, or far from each other–they are on other objects. If the points lie on a straight line parallel to the plane of the cameras, the difference between successive matched pixels is the same; if the line is not parallel, but the points belong to the same object, this difference should not vary significantly, and in the case where the points do not belong to the same object, the difference between neighboring matches is greater and depends on the distance between the objects. The choice of such a gap penalty is intended to minimize the problem of many false matches in areas of uniform color where the difference in pixel color is small, but even careful selection of the value of the *d* parameter does not eliminate all such false matches; the algorithm always matches better where the color differences are larger. The design of the algorithm using only the color of the pixels also causes other false matches to occur due to the fact that the corresponding pixels differ in color, e.g., due to light reflected by the object, which is observed from other points in space. Further false matches may result from the fact that the algorithm may miss narrow objects, due to gap penalties; avoiding two gaps may give a larger *F* function value than a correct match.

Equation (Equation 3) contains the definition of the recursive formula used in the stereo vision-adapted version of the Needleman–Wunsch algorithm.
(3)$$F\left(i,j\right)=\max\left\{\begin{array}{c} F(i-1,j-1)+e(i,j)\\ F(i-1,j)+d(n)\\ F(i,j-1)+d(n) \end{array}\right.$$

In this formula, F(i,j) denotes the value in the *i*-th row and *j*-th column of the auxiliary matrix, e(i,j) is the pixel match—the reward (penalty) for the match, and d(n) denotes the gap penalty function. The best match is obtained by finding the path from the last cell of the auxiliary matrix (from the bottom right corner) to its first cell (top left corner), always choosing the action whose value is the largest.

For example, if we match lines (sequence of pixels) shown on Figure 2, where we denote the first line as $s=s_{0}s_{1}s_{2}s_{3}s_{4}s_{5}$, and second line as $t=t_{0}t_{1}t_{2}t_{3}t_{4}t_{5}$ (in the example, we assume lines on an image have 6 pixels), and we denote colors as A, B and C, therefore s=AABCCC, t=ABBBBC, and use the example values of similarity from Equation (Equation 1) $e\left(i,j\right)=\begin{array}{cccc} & A & B & C\\ A & 11 & -9 & -1\\ B & -9 & 8 & -14\\ C & -1 & -14 & 7 \end{array},$ and the penalty d=−5, the results is $\begin{array}{cccccccc} A & - & - & A & B & C & C & C\\ | & & & & | & | & & \\ A & B & B & B & B & C & - & - \end{array}$ or matched pixels are $\left(s_{0},t_{0}\right),\left(s_{1},t_{3}\right),\left(s_{2},t_{4}\right)$. The F(i,j) from Equation (Equation 3), and results, are depicted in Figure 2.

Figure 2 shows the auxiliary matrix of the algorithm with the matches and gaps marked. The gaps have an additional indication of which image they are visible in.

Matching sequences with the Needleman–Wunsch algorithm requires filling the auxiliary matrix, that is, completing all its $N^{2}$ cells, and finding a path from the lower left to the upper right corner of the matrix, which requires *N* operations. The computational complexity of the algorithm is thus $O(N^{2})$.

#### 2.1.1. Disparity Map Calculation Based on Matching

Adaptation to the problem of finding matching pixels on a pair of images also required the second step of the algorithm. In our implementation, when finding the best path in the filled auxiliary matrix, a disparity map (an image containing the distances between images of the same point on the plane of the two cameras) is completed based on the coordinates of the cells corresponding to the matched pixels. For each match, a value of $x_{r}-x_{l}$ or $x_{l}-x_{r}$ is stored in the disparity map, where $x_{l}$ and $x_{r}$ are the coordinates of the matched pixels in the left and right images.

The disparity map calculated in this way contains only the values for those pixels for which matching has happened. To obtain a complete disparity map, it is necessary to fill in the holes that appear. We mentioned in Section 2.1 that the gaps are due to the fact that a given section of space is visible from only one camera.

Figure 3 shows a series of images with marked gaps relative to the centre (reference) image, and Figure 4 shows the gaps calculated using the algorithm from our Stereo PCD library without filling in the gaps.

For this reason, we fill in the holes based on the value of the pixel neighboring the gap, corresponding to the portion of space further away from the camera plane.

#### 2.1.2. Improving the Quality of Matching through Edge Detection

The first improvement we propose is due to the desire to use additional information, aside from color, on which to compare pixels. The improvement we propose uses the Harris feature and edge detector [26]. It is a fast corner and edge detector based on a local autocorrelation function. It works with good quality on natural images.

In our solution, we use the information, about edges, to add a reward for matching if both related pixels are on an edge.

Another rather simple but, as it turned out, effective improvement relates to the filling of gaps in disparity maps. We propose to use both neighbors of the gap and information about any pixels recognized as edges and inside the hole to fill the gap. The value of the neighbor previously unused in filling the hole is now used to fill it up to the first pixel recognized as an edge, while the rest of the hole is filled as before by the other neighbor.

**Figure 4 sensors-24-05786-f004:**
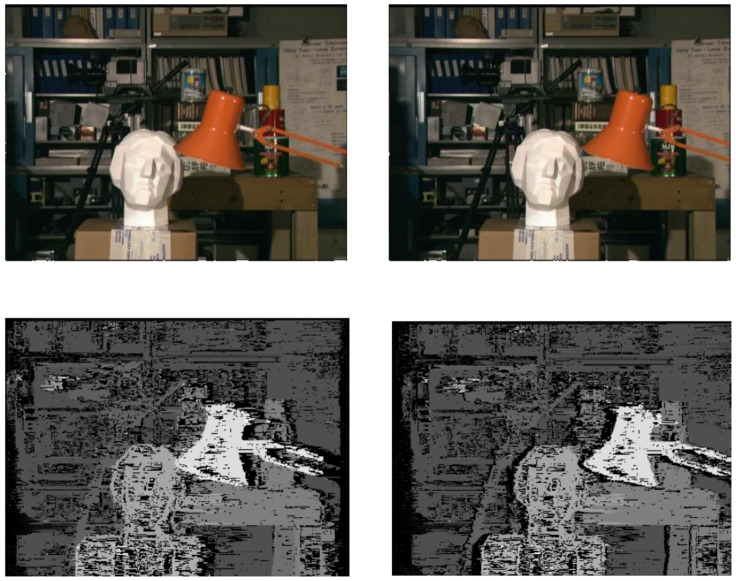
Images from the ‘head and lamp’ dataset [27] along with the determined disparity map for the left and right images without filling in the gaps.

#### 2.1.3. Performance Improvement, Reducing Length of Matched Sequences

We have also applied improvements to produce a result in less time. The first improvement takes advantage of the fact that it is not necessary to compare all pixels from a given line from one image with all pixels from the corresponding line from the other image. This is because, knowing the parameters of the measurement set, it is possible to determine at what minimum distance the object is from the camera and, therefore, to determine the maximum value of the disparity. In addition, knowing which image contains the view from the left and which from the right camera allows you to consider that the minimum disparity is 0. The use of these two facts allows you to reduce the number of pixels compared with each other. Most often, a single pixel from a single image only needs to be compared with several times fewer pixels than when whole lines are compared. Thus, the application of this improvement improves the computational complexity of the pixel-matching algorithm on a pair of images. The original complexity is $O(HW^{2})$, where *H* is the height of the image and *W* is the width of the image—this is due to the need to fill a square matrix with $W^{2}$ cells for each *H* line. Using the disparity constraint reduces the computational complexity to O(HWD), where *D* is the maximum value of the disparity (smaller than the image width *W*). The improvement is due to the need to fill in at most *D* (instead of *W*) cells in *W* rows of the auxiliary matrix. Moreover, taking advantage of this fact, we decided to write this array in such a way that it takes up less memory. Instead of allocating an array of size W×W, we only need an array of size W×D, so that the number of columns is equal to the maximum number of comparisons for a single pixel. Applying such a trick, however, required changing the indexes of the cells used to calculate the value to be written in the next cell of the array—the indexes of the cells read from the row above the complement increased by 1. This is shown in Figure 5.

#### 2.1.4. Parallel Algorithm for 2D Images Matching

The next improvement takes advantage of the fact that lines (rows of pixels) are matched independently of each other. This allows you to analyze several lines in parallel. For this purpose, we use multithreading in our solution. Each thread allocates its buffers for data, e.g., the auxiliary matrix used in the algorithm, and then takes the value of the line counter and increments its value. Synchronization is ensured so that two threads do not fetch the same line number. The thread then matches the pixels on that line, fetches the next line number, matches the pixels on that line, and the process repeats until the calculations for all lines are complete.

## 3. Results

The algorithm presented in our work has been implemented in C++ and placed in the Stereo PCD open library, along with other tools supporting the process of combining camera and lidar data. The library is accessible from the Python language level, and the algorithm implemented in C++ was placed in it thanks to the binding. The algorithm from the Stereo PCD library was tested on publicly available datasets that are frequently used for evaluating the quality of stereo vision algorithms.

### 3.1. Datasets

To evaluate the quality and performance of the algorithm used to find pixel matches on a pair of stereo vision images, we used open and online datasets. We chose such datasets so that you can easily compare with other solutions. Additional motivation for selecting a particular dataset is described at the end of the paragraph describing that dataset. The number of algorithm rankings on the [28,29] indicates the popularity of the selected datasets.

#### 3.1.1. University of Tsukuba ‘Head and Lamp’

The ‘head and lamp’ dataset [27] consists of images showing a view of the same scene showing, among other things, a head statue and a lamp. The dataset is made up of low-resolution images, which allows the disparity to be calculated fairly quickly even with a low-performance algorithm, which is very useful in the development and initial testing stages. The dataset is available at [28]. See Figure 6.

#### 3.1.2. Middlebury 2021 Mobile Dataset

The Middlebury 2021 Mobile dataset [28,30] was created by Guanghan Pan, Tiansheng Sun, Toby Weed and Daniel Scharstein in 2019–2021. The dataset contains high-resolution images that were taken indoors using a smartphone mounted on a robotic arm. We chose this dataset because it contains actual high-resolution data, which is what modern cameras provide; additionally, thanks to the scene created to take the images, they contain a high-quality reference disparity map. An additional motivation was that they contain images taken indoors, so we can evaluate how the algorithms perform on data collected in such an environment. The dataset is available at [28]. See Figure 7.

#### 3.1.3. KITTI

The KITTI dataset [29,31] was developed by the Karlsruhe Institute of Technology, Germany and the Toyota Technological Institute at Chicago, IL, USA and published in 2012. The data were collected from sensors placed on the moving car, including stereo cameras and lidar. Based on the data collected over 6 h of driving, datasets were created to test algorithms including object detection and tracking and depth estimation based on image pairs. We chose this dataset due to the fact that it contains real road data, so we could evaluate the algorithms in the context of application to autonomous vehicles. The datasets are available at [29]. See Figure 8.

### 3.2. Quality Evaluation Method

In order to evaluate the effects of matching and to select the parameters of the algorithm, it is necessary to be able to estimate the quality of the matching and to be able to assess which matching is better. A typical approach to this type of issue is the calculation of error statistics. Such statistics are usually calculated by comparing the obtained result with a pattern from the provided dataset.

Quite popular metrics for this problem used in [32], among others, are the root of the mean square error defined by the Equation (Equation 4) and the percentage of correct matches calculated from Equation (Equation 5),
(4)$$RMSE=\sqrt{\frac{1}{N}\sum_{\left(x,y\right)}{\left(d_{C}\left(x,y\right)-d_{G}\left(x,y\right)\right)}^{2}}$$
(5)$$GPR=\frac{1}{N}\sum_{\left(x,y\right)}\left(\right|d_{C}\left(x,y\right)-d_{G}\left(x,y\right)\left|\;<\;\delta_{d}\right)$$
where $d_{C}$ is the calculated disparity value for a given pixel, $d_{G}$ is the benchmark disparity value for a given pixel, *N* is the number of pixels evaluated, and $\delta_{d}$ is the acceptable matching error. If the error is less than $\delta_{d}$, the pixel is considered a correct match; otherwise, it is a false match.

Since there are several ways to calculate quality metrics, we decided to implement a function to calculate such quality metrics in the Stereo PCD library. The library implements evaluation methods with skipping pixels without disparity values and considering pixels without values in the calculated disparity maps as bad matches.

### 3.3. Quality Tests

We tested the quality of the implemented algorithm using data from the KITTI and Middlebury 2021 Mobile dataset described in this section. We used a package from the Stereo PCD library to assess the quality. Furthermore, we noticed that for the results for the data from both datasets, the error statistics differed significantly between image pairs, even when it came to image pairs from the same dataset. We noticed that those pairs of images on which the algorithm achieves good results are those with varied colors, containing a lot of objects far from the camera, depicting equally lit areas, without shadows and large areas of uniform color. In contrast, the algorithm performed poorly if the images contained areas of uniform color (becoming worse the closer such areas were to the observer), and on images where there were light reflections (e.g., reflections of artificial light in images from the Mobile dataset or reflections of light from car windows in images in the KITTI dataset). Therefore, we decided to present these results, in addition to tabulating the average error values, in the form of histograms showing the number of image pairs for which the error statistics were within a given range.

We ran the tests for a number of parameter values and discuss the results for the best set of parameters found; for all images from the set, we use the same parameter values. The parameters used during the quality tests are $e_{max}=0.05$, $d_{start}=-0.015$, $d_{continue}=-0.005$ and an edge detector parameter of 0.0001 for the KITTI dataset and $e_{max}=0.02$, $d_{start}=-0.015$, $d_{continue}=-0.08$ and an edge detector parameter of 0.0001 for the Mobile dataset. We ran the algorithm on images in YUV color space.

We performed a similar experiment, using the same datasets as for our algorithm, for the SGBM [33] (Semiglobal Block Matching) algorithm from the OpenCV library, in which pixel matching is based on a smoothness constraint, which is usually expressed as a global cost function. SGBM performs fast approximation by optimizing paths from all directions. We ran the SGBM algorithm for grayscale images with the parameter set described on the website for the KITTI dataset [29] and almost the same set for the Mobile dataset; we only changed the value of the numDisparities parameter to 512.

In addition, because the SGBM algorithm does not compute a dense disparity map, but leaves pixels without values, we tested it in two ways—one in which pixels without values in the computed disparity map are not taken into account when calculating error statistics and another in which such pixels are treated as false matches.

Tests on the KITTI set have shown that the implemented algorithm, with the same set of parameters, can achieve results that definitely differ in quality level. By increasing the tolerable error threshold, the number of pixels considered to be correctly matched increases, and the differences between the ratio of correctly matched pixels for different image pairs are minimally smaller. This is visible through the data in the Table 1.

The version using the edge detection improvement described in Section 3.2 achieved better results. Observing the histograms in Figure 9, it can be seen that there are few image pairs for which the algorithm achieved a very poor result.

In comparison, the SGBM algorithm for image pairs from the KITTI set achieves high results. Histograms showing the error statistics of pixel matches from a stereoscopic pair computed using the SGBM algorithm are in Figure 10.

Increasing the tolerable error threshold in the range for most pairs of images gives a significantly higher result; only for a few pairs of images, the rate of correct matches increases little after raising the error tolerance threshold. This is observable for the data in the Table 1 and the graphs in Figure 11.

The algorithm from the Stereo PCD library achieves worse results for the 2021 Mobile dataset than for the KITTI dataset. In the case of images from this set, it is also evident that the quality of the result strongly depends on the image pair—this is shown, among other things, by the histograms in Figure 12 and the data in Table 2. Similarly, as with the results for the KITTI dataset, it can be seen that there are few image pairs for which the algorithm scored very poorly.

For the 2021 Mobile dataset, the SGBM algorithm also performs worse in quality tests than the KITTI dataset. Histograms showing the ratio of correctly matched pixels for the data from this set can be found in Figure 13. As for the data from the KITTI set, increasing the threshold of acceptable error caused a noticeable increase in the value of the ratio of correctly matched pixels.

There are a lot of pixels without values on the calculated disparity maps—on average, about 44% of the pixels from the reference disparity map do not have their correspondents on the calculated disparity map. Recognizing such pixels as false matches results in significantly lower values of the rate of correct matches. Note, however, that this result can be easily improved by adding a step after executing the algorithm to fill the gaps in the calculated disparity maps, for example, using the values of pixels neighboring the gap.

In summary, the SGBM algorithm from the OpenCV library achieves better quality results than the algorithm from the Stereo PCD library. Moreover, the quality of the results for our algorithm definitely depends more on the input image pair. In the case of SGBM, there is also a similar dependence, but the differences in quality scores are much smaller.

### 3.4. Method of Performance Evaluation

To evaluate algorithms, one uses not only measures of the quality of the results obtained but also, among other things, the time it takes to obtain it. Therefore, we also ran performance tests of the pixel-matching algorithm on a pair of stereoscopic images. We tested the processing time of the image pair using the time function from the time module of the standard Python v.3.10 language library. We ran the tests on a personal computer; the results may vary depending on the parameters of the machine on which they were run. Nonetheless, it allows evaluating which algorithm runs faster and how the resolution of the images affects the running time of the algorithms. To make the results reliable, we ran the tests for different images from the collections and repeated them several times.

### 3.5. Performance Tests

The achieved results are presented in Table 3. In it, we compared two versions of the algorithm from the Stereo PCD library—a multithreaded one with all the proposed improvements from Section 2.1.2 and Section 2.1.3 and without the edge-related improvement, and the SGBM algorithm from the OpenCV library. The running times of the algorithms and GPR shown in the table were measured with and without the maximum disparity parameter set to 200 pixels.

The use of a constraint on the maximum value of disparity has resulted in similar matching times for images of the same resolution but with different orientations. This improvement also improves run times by several times.

Significant speed-up was also achieved by using a multithreaded version of the algorithm. Our Stereo PCD is faster than OpenCV algorithm, as depicted in Table 3.

## 4. Discussion

The goal of our research was to apply a method used in bioinformatics for aligning sequences of amino acids to the task of matching stereo vision images. The experiments we conducted on publicly available datasets demonstrated the effectiveness and potential of this method for stereo vision tasks. By setting appropriate parameter values for the break penalty value, we can obtain an accurate and smooth disparity map. Due to the fact that the lines are matched independently, the obtained solutions are not optimal globally but rather only optimal inside one pair of scanlines. For this reason, there are situations when the values of the disparity in neighboring lines significantly deviate from each other, despite the fact that they occupy a very similar fragment of the space.

Our algorithm needs a deterministic number of operations; it is not supported by statistical models like machine learning (ML) models. Further work could explore the opportunity of using ML models to speed up calculations for typical images.

In the next step of work on the presented algorithm, pixel matches from neighboring lines could be used so that the algorithm can force smoothness of the disparity in more than two directions, e.g., smoothness of a pixel in the direction of all its eight neighbors. This would require modifying the algorithm and proposing a matching function analogous to that for single-scanline matching. Such a solution will indeed increase the computational complexity of the algorithm but will potentially achieve higher-quality solutions.

Another potential way to improve the algorithm could be to try to speed up line matching. One potential way to achieve this is to parallelize the computation. Here, GPUs could be used, and the dynamic programming algorithm could be implemented in a vector-based approach—simultaneously completing the cells of the auxiliary matrix located on the same antidiagonal. An alternative way to speed up the algorithm is to reduce the number of pixels comparing each other when matching lines. For this purpose, matches from neighboring lines can be used, and the search space for a single pixel can be adaptively limited with a full search applied only every few lines. Such an improvement of the algorithm will allow reducing CPU consumption while matching the same number of frames per second as in the presented version, and thus application on small devices, i.e., embedded systems or mobile devices. Reducing the number of operations of the algorithm also makes it possible to increase the number of processed frames per second without changing the hardware, which could allow the algorithm to be applied to devices in motion, e.g., cars and real-time processing; depending on what speed-up is achieved, the algorithm could be used, for example, not only on roads in the city but also on routes between cities.

In our opinion, the proposed algorithm can be used as a basis for creating more and better, both faster and more accurate, methods for fusing stereo-camera and lidar data.

## 5. Summary

This paper presented a new algorithm for point cloud densification for fusing lidar and camera data. This algorithm was provided in the new open-source library named Stereo PCD, which was designed for Python 3. The library’s source code consists of about 1700 lines of code, of which 1000 lines are in Python and 700 lines are in C++. It enables the processing of stereo image pairs and point clouds, e.g.,
The creation of point clouds based on pairs of stereo vision images and camera calibration data;Combining several point clouds together;Coloring of lidar point clouds based on camera images;Saving point clouds containing point coordinates or coordinates with color in formats that allow loading into a visualization tool MeshLab [34] or an object detector in a point cloud in three-dimensional space, e.g., OpenPCDet [35].

Moreover, the library includes a number of functions for determining extrinsic and intrinsic calibration matrices, projection matrices or distances between cameras based on various parameters.

The library provides support for the Middlebury 2021 Mobile dataset [30], mentioned in Section 3.1.2, and the KITTI Stereo dataset [31], mentioned in Section 3.1.3.

The results show that our library performs better than OpenCV and has comparable quality.

## Figures and Tables

**Figure 1 sensors-24-05786-f001:**
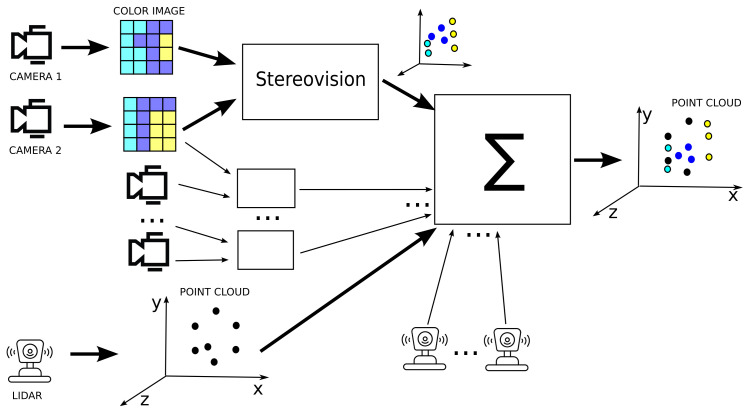
Overview of how to fuse multi-camera and lidar data. A 3D scene is reconstructed from the stereo camera images, and a color point cloud is created. Such point clouds are combined with the point clouds from multiple lidars into a single point cloud. The resulting point cloud has lidar points, depicted in black, and stereovision output (presented in colour).

**Figure 2 sensors-24-05786-f002:**
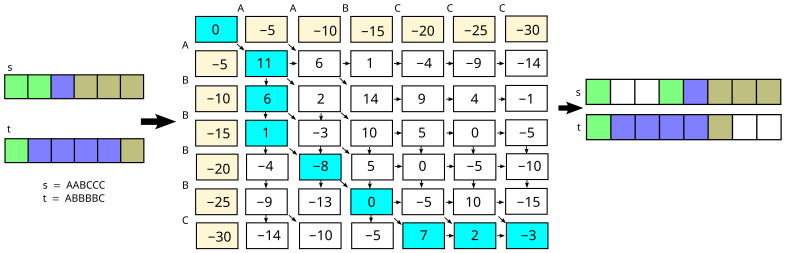
The matching algorithm, example, s,t are sequences of green, blue and yellow pixels, denoted as A, B, and C respectively (on the left), the Needleman–Wunsch matrix in the middle (result is depicted in sky-blue), the matching results on the right. The parameters are depicted in the text.

**Figure 3 sensors-24-05786-f003:**
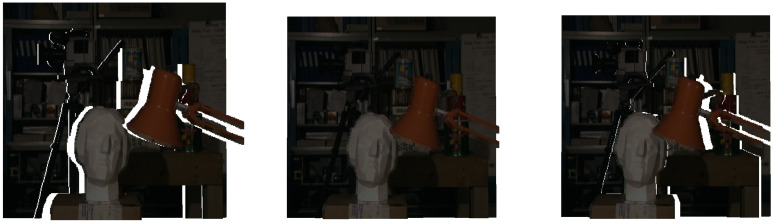
A series of images showing a view of the same scene. The reference image is in the middle. The gaps are highlighted in white.

**Figure 5 sensors-24-05786-f005:**
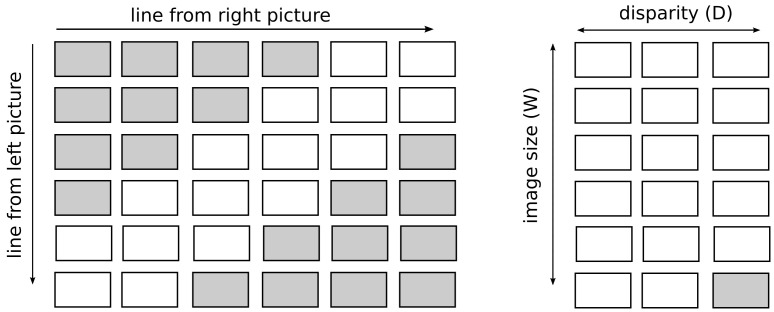
The auxiliary matrix of the algorithm after the changes. The gray color indicates those cells that will not be filled in. On the left, coordinates of pixels on the left and right image, on the right, the reduced matrix that considers disparity.

**Figure 6 sensors-24-05786-f006:**
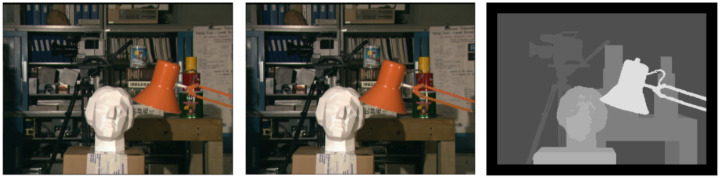
Examples of images from the ‘head and lamp’ dataset with a reference disparity map.

**Figure 7 sensors-24-05786-f007:**
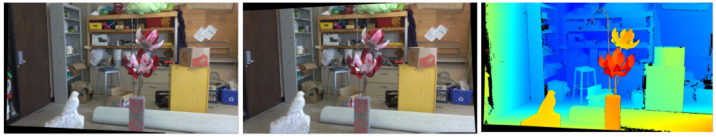
A pair ‘artroom1’ images from the Middlebury 2021 mobile dataset along with a reference disparity map.

**Figure 8 sensors-24-05786-f008:**
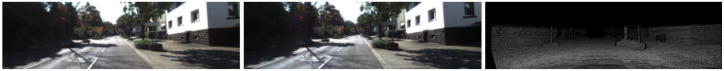
Examples of images from the KITTI dataset with a reference disparity map.

**Figure 9 sensors-24-05786-f009:**
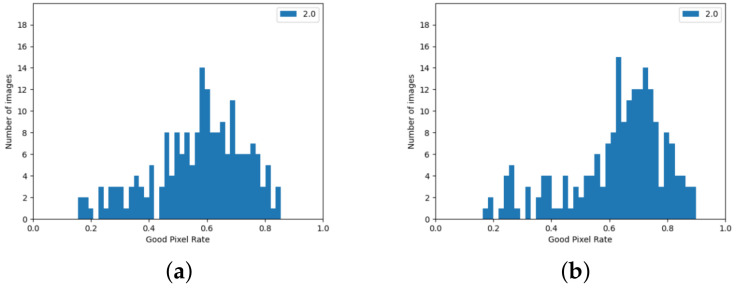
GPR value for images from the KITTI set, achieved using algorithm from Stereo PCD. (**a**) The version that does not use edge information, (**b**) The version using edge information.

**Figure 10 sensors-24-05786-f010:**
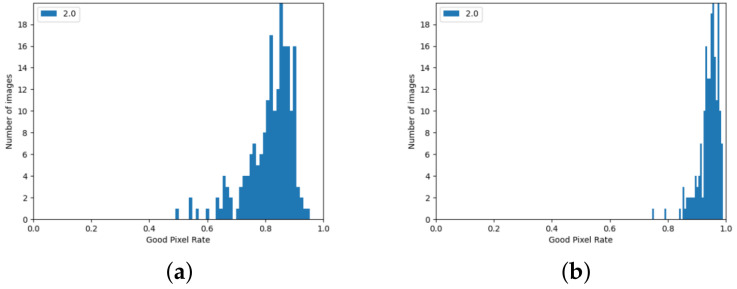
GPR values for images from the KITTI set achieved using the SGBM algorithm. (**a**) Values calculated when pixels with no value are considered a matching error, (**b**) calculated values when pixels without disparity values are omitted.

**Figure 11 sensors-24-05786-f011:**
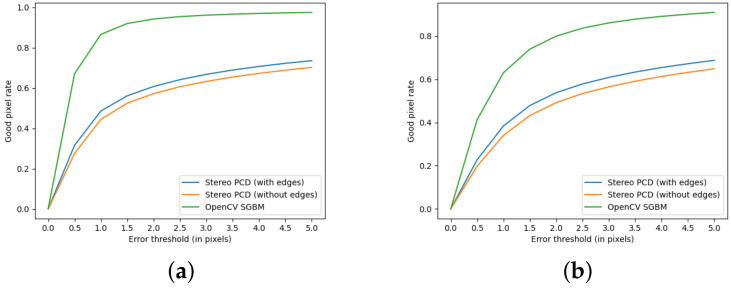
Good pixels rate as function of error threshold for images for KITTI and Mobile datasets. (**a**) Values for the KITTI dataset, (**b**) values for the Mobile dataset.

**Figure 12 sensors-24-05786-f012:**
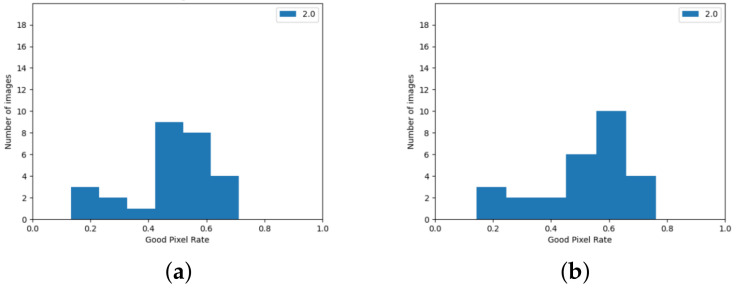
GPR values for images from the 2021 Mobile dataset achieved using from Stereo PCD. (**a**) The version that does not use edge information, (**b**) the version using edge information.

**Figure 13 sensors-24-05786-f013:**
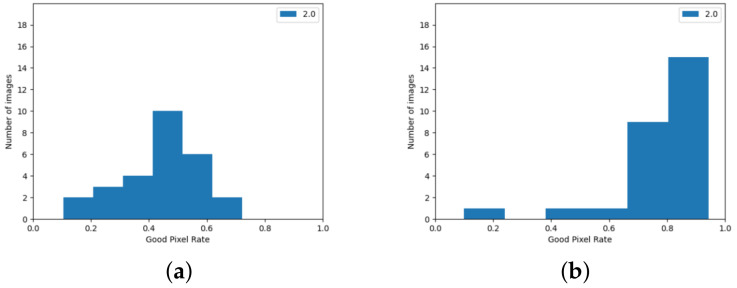
GPR values for images from the 2021 Mobile dataset achieved using the SGBM algorithm. (**a**) Values calculated when pixels with no value are considered a matching error, (**b**) calculated values when pixels without disparity values are omitted.

**Table 1 sensors-24-05786-t001:** Averaged quality results of different algorithm variants in finding a match for images from the KITTI dataset. The table includes RMSE—root-mean-square error and GPR—good pixel rate. The false match rate is approximately 1−GPR.

Algorithm	RMSE	${\mathrm{GPR}}_{1}$	${\mathrm{GPR}}_{2}$	${\mathrm{GPR}}_{4}$
Stereo PCD (with edges)	15.91 ± 11.77	0.49 ± 0.16	0.61 ± 0.16	0.71 ± 0.16
Stereo PCD (without edges)	17.12 ± 10.07	0.45 ± 0.16	0.57 ± 0.16	0.67 ± 0.15
OpenCV SGBM (all)	3.31 ± 2.56	0.75 ± 0.09	0.81 ± 0.08	0.84 ± 0.07
OpenCV SGBM (valid)	3.31 ± 2.56	0.86 ± 0.06	0.94 ± 0.04	0.94 ± 0.02

**Table 2 sensors-24-05786-t002:** Averaged quality results of different algorithm variants in finding a match for images from the Middlebury 2021 Mobile dataset.

Algorithm	RMSE	${\mathrm{GPR}}_{1}$	${\mathrm{GPR}}_{2}$	${\mathrm{GPR}}_{4}$
Stereo PCD (with edges)	22.94 ± 11.40	0.38 ± 0.13	0.54 ± 0.14	0.65 ± 0.14
Stereo PCD (without edges)	24.79 ± 12.74	0.34 ± 0.12	0.49 ± 0.13	0.61 ± 0.13
OpenCV SGBM (all)	13.54 ± 6.02	0.36 ± 0.12	0.45 ± 0.13	0.50 ± 0.13
OpenCV SGBM (valid)	13.54 ± 6.02	0.63 ± 0.12	0.80 ± 0.10	0.89 ± 0.06

**Table 3 sensors-24-05786-t003:** Averaged running time and GPR values for the 2-pixel threshold obtained for different variants of the match-finding algorithm for images of different resolutions from different datasets.

Dataset	Image Resolution	Algorithm	Time	${\mathrm{GPR}}_{2}$
		Stereo PCD	0.02 s	0.90
Head and lamp	384×288	Stereo PCD (200 px)	0.02 s	0.90
		OpenCV SGBM	0.03 s	0.94
		Stereo PCD	0.33 s	0.57
		Stereo PCD (200 px)	0.15 s	0.57
KITTI	≈1240 ×375	Stereo PCD (edges)	0.40 s	0.61
		Stereo PCD (edges + 200 px)	0.17 s	0.63
		OpenCV SGBM	0.20 s	0.94
		Stereo PCD	0.79 s	0.54
		Stereo PCD (200 px)	0.51 s	0.54
Mobile dataset	1080×1920	Stereo PCD (edges)	1.80 s	0.58
		Stereo PCD (edges + 200 px)	0.74 s	0.58
		OpenCV SGBM	1.22 s	0.75
		Stereo PCD	2.04 s	0.49
		Stereo PCD (200 px)	0.58 s	0.45
Mobile dataset	1920×1080	Stereo PCD (edges)	4.22 s	0.53
		Stereo PCD (edges + 200 px)	0.86 s	0.49
		OpenCV SGBM	1.72 s	0.81

## Data Availability

Stereo-PCD is available on the https://github.com/jwinter3/Stereo-PCD (accessed on 2 September 2024) repository under the MIT licence.

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
