# Peer review of "Point Cloud Densification Algorithm for Multiple Cameras and Lidars Data Fusion"

_sensors, 2024, doi:10.3390/s24175786_

Round 1
Reviewer 1 Report
Comments and Suggestions for Authors
The authors propose a dynamic programming and the adaptation of the Needleman-Wunsch algorithm for pixel matching to fuse data from multiple cameras with data from multiple lidar. This paper is well-written and easy to follow.
I have three main suggestions for improvement:
1) The literature review could be more comprehensive. While it mentions some related work, it lacks depth in discussing the various existing methods and their limitations, which the proposed method aims to address.
2) The experimental setup and parameters used for benchmarking are not described in sufficient detail. Including these details would make it easier to replicate the experiments.
3) The paper does not discuss potential edge cases or limitations of the proposed method. Addressing these aspects would provide a more balanced view of the algorithm’s applicability and robustness.
Comments on the Quality of English LanguageThe English languague is generally good, but some minor revision is needed. For example, line 120:
Assuming a pinhole camera model, considering two cameras producing stereoscopic images, to matching image of pixels (2D array) we can properly match sequences,
should be changed to
Assuming a pinhole camera model, considering two cameras producing stereoscopic images, to match image of pixels (2D array), we can properly match sequences.
Please check the whole manuscript to aviod such grammatical problem.
Author Response
Thank you for constructive feedback. Our response is in attached file.

Reviewer 2 Report
Comments and Suggestions for Authors
1. While the abstract is well written, it refers to motivation in the autonomous vehicles, a topic that is not adequately covered in the text, and this is confusing.
2. While the introduction is informative, it should be better organised to separate motivation from literature findings.
3. Line 166: The situations that may cause false matches should be elaborated.
4. Lines 257-258: Main criteria for selection of datasets must be elaborated.
5. Lines 289-290: Situations for false matching should be defined.
6. Line 299: Please define how the difference is a function of the image characteristics.
7. Table 1 should also summarize false matching.
8. Lines 326-329: Would be useful to present performance indicators as function of error threshold -- as established in the literature.
9. Table 3 should combine together the three indicators, i.e. successful matching, false matching and time. For example, it is not clear how the improvement of line 373 affects the other indicators.
10. Result on line 429 is too general, please specify.
Comments on the Quality of English Language
A complete check of English language is needed. For example, see lines 42, 119, 197, 365, 428, 436.
Author Response

(The authors gave the same response as above.)

Round 2
Reviewer 2 Report
Comments and Suggestions for Authors
Thank you for the improvements.
Comments on the Quality of English LanguageThank you for the improvements.